# Regional Splanchnic Oxygenation during Continuous versus Bolus Feeding among Stable Preterm Infants

**DOI:** 10.3390/children9050691

**Published:** 2022-05-09

**Authors:** Gisela Laura Sirota, Ita Litmanovitz, Carmel Vider, Shmuel Arnon, Shiran Sara Moore, Eynit Grinblatt, Orly Levkovitz, Sofia Bauer Rusek

**Affiliations:** 1Department of Neonatology, Meir Medical Center, Kfar-Saba 4428164, Israel; litmani@clalit.org.il (I.L.); shmuelar@clalit.org.il (S.A.); p.shiran@gmail.com (S.S.M.); sheynitgr@clalit.org.il (E.G.); orly.levko@gmail.com (O.L.); bauers@clalit.org.il (S.B.R.); 2Sackler Faculty of Medicine, Tel Aviv University, Tel Aviv 6997801, Israel; 3Department of Pediatrics, Meir Medical Center, Kfar-Saba 4428164, Israel; carmel.vider@gmail.com

**Keywords:** preterm, NIRS, splanchnic regional oxygenation

## Abstract

Introduction: There is no agreement regarding the best method for tube-feeding preterm infants. Few studies, to date, have evaluated the influence of different methods of enteral feeding on intestinal oxygenation. The use of near-infrared spectroscopy (NIRS) has permitted the noninvasive measurement of splanchnic regional oxygenation (rSO_2_S) in different clinical conditions. The aim of this prospective, single-center study was to compare rSO_2_S during continuous versus bolus feeding among stable preterm infants. Methods: Twenty-one preterm infants, less than 32 weeks gestation and appropriate for gestational age, were enrolled. All infants were clinically stable and on full tube feedings. Each infant received a bolus feeding initially (20 min duration), and after 3 h, a continuous feeding (5 h duration). Infants were evaluated 30 min before and 30 min after the bolus and continuous feedings. The regional splanchnic saturation (rSO_2_S) was measured using near-infrared spectroscopy (NIRS) technology and systemic saturation was measured with pulse oximetry. From these measurements, we calculated the splanchnic fractional oxygen extraction ratio (FOES) for each of the four intervals. Results: rSO_2_S decreased after continuous vs. bolus feeding (*p* = 0.025), while there was a trend toward decreased SaO_2_ after bolus feeding (*p* = 0.055). The FOES, which reflects intestinal oxygen extraction, was not affected by the feeding mode (*p* = 0.129). Discussion/Conclusion: Continuous vs. bolus feeding decreases rSO_2_S but does not affect oxygen extraction by intestinal tissue; after bolus feeding there was a trend towards decreased systemic saturation.

## 1. Introduction

Optimal enteral feeding of premature infants has a strong impact on their short- and long-term outcomes. It is widely accepted that human milk is the optimal nutrition for preterm infants, and it is associated with decreased incidences of sepsis and necrotizing enterocolitis (NEC) [1]. There is also enough evidence that early initiation of enteral feeding decreases metabolic and infectious complications and improves intestinal function [2,3,4]. However, there is no agreement on the best method for tube feeding [2,3]. Studies that compared intermittent bolus feeding to continuous feeding reported inconsistent results regarding growth, time to achieve full enteral feeding, and gastrointestinal complications [4,5].

Previous studies evaluated the association between intestinal blood flow and gastrointestinal morbidities [6,7]. Fang et al., assessed the mesenteric perfusion in preterm infants with Doppler and found a significant correlation between increased blood flow velocity in the superior mesenteric artery and an early tolerance to enteral feeding [6].

In the last decade, the use of near-infrared spectroscopy (NIRS (technology for the assessment of mesenteric perfusion has been increasing [8,9]. NIRS is based on the principle that most biological tissues, other than hemoglobin and cytochrome oxidase, are relatively transparent to infrared light in the 700–1000 nanometer range. The absorbance spectrum of hemoglobin depends on its oxygenation status: deoxygenated hemoglobin absorbs more red light and less infrared light compared to oxygenated hemoglobin. The device emits light at wavelengths within the above-mentioned spectral range and analyzes photons returning to the transducer. The change in the intensity of the reflected light depends on the oxyhemoglobin-to-deoxyhemoglobin ratio, from which oxyhemoglobin saturation in the underlying tissue can be derived [10].

The measurement of splanchnic regional oxygenation (rSO_2_S) by NIRS reflects mesenteric perfusion and oxygenation [8,9], with a good correlation between rSO_2_S and superior mesenteric artery blood flow, as measured by Doppler [11].

The association between splanchnic oxygenation and feeding intolerance in infants was studied previously. In 2015, Dani et al., reported that rSO_2_S, when measured during the first days of life, does not predict the time it takes to achieve full enteral feeding [12]. In contrast, Corvalia et al., found that lower regional oxygenation before and after the first feeding predicts feeding intolerance [13].

Several studies investigated the impact of enteral feeding on splanchnic oxygenation in preterm infants [9,11,12,13,14,15], but to our knowledge, only two small studies have compared the influence of different methods of enteral feeding on intestinal oxygenation [14,15]. Although an increase in rSO_2_S following bolus feeding was found, there were inconsistencies regarding the effects of continuous feeding. Dani et al., found no effect on intestinal oxygenation [14], while Corvalia et al., showed decreased oxygenation following continuous feeding [15].

The current study compared rSO_2_S and oxygen extraction by intestinal tissue during continuous vs. bolus feeding among stable, preterm infants.

## 2. Materials and Methods

Preterm infants born at Meir Medical Center, Kfar Saba, Israel, participated in the study. Infants were eligible if they met the criteria of a gestational age <32 weeks, weight appropriate for their gestational age, clinically stable at the time of the study, on full enteral feedings (150 mL/kg/day) for at least a week before enrollment, and tolerated bolus feeding the day before the study (every 2 or 3 h according to the local enteral feeding protocol; gastric residuals <25% of the feeding volume). 

Premature infants with intrauterine growth retardation, major congenital anomalies, severe central nervous system disorder, gastrointestinal disorder, cutaneous disease, prior diagnosis of NEC or spontaneous intestinal perforation, and those who received blood product transfusion or vasopressor therapy up to a week before enrollment, were excluded from the study.

### 2.1. Study Protocol

Each patient was first given a bolus feeding (20–30 min duration), and after 2 or 3 h started 5 h of continuous feeding of the same type of milk (fortified human milk or preterm formula). The bolus feeding regimen was then restored.

The volume of the bolus feed was the same as that given the day before the study, while the volume of the continuous feed was calculated by dividing the total daily requirements into 4 meals (25%). It was administered over a period of 5 h, followed by an hour pause.

During the study period, infants were kept in the supine position with minimal handling to minimize NIRS artifacts. The arterial oxygen saturation (SaO_2_) was continuously measured using pulse oximetry (Massimo). The rSO_2_S was measured with NIRS (INVOS 5100, Somanetics Corporation, Troy, MI, USA), using a sensor over the splanchnic bed, placed just below the umbilicus, according to the manufacturer’s recommendations (Figure 1). SaO_2_ and rSO_2_S were measured continuously from 30 min before the bolus feeding to 30 min after the continuous feeding was completed.

The study was divided into 4 intervals: Period 0—30 min before the bolus feeding, Period 1—30 min after the bolus feeding, Period 2—30 min before the continuous feeding, and Period 3—30 min after the continuous feeding. For each period, the average SaO_2_, rSO_2_S, and FOES values were calculated.

The FOES was calculated as the difference between arterial SaO_2_, measured with pulse oximetry, and rSO_2_S, measured by NIRS (FOES = (SaO_2_‒rSO_2_S)/SaO_2_). This parameter reflects the balance between oxygen delivery and oxygen consumption. Therefore, an increase in the FOES suggests an increase in tissue oxygen extraction due to higher oxygen consumption in relation to oxygen delivery, while a decrease suggests less oxygen use in comparison with the supply.

Infants’ characteristics were recorded from their electronic medical records, including gestational age, birth weight, sex, type of delivery, age and weight at study entry, type of feeding, age at full enteral feeding, and prematurity-related complications (respiratory distress syndrome, need of surfactant, patent ductus arteriosus, intraventricular hemorrhage, NEC, or bronchopulmonary dysplasia).

### 2.2. Statistical Analysis

We calculated that a sample size of at least 12 infants was required. The power calculation was based on the results of Dani et al., to detect a statistically significant difference of 10% in rSO_2_S, 30 min after bolus feeding, with 80% power at 0.05 levels [14]. SaO_2_2 and rSO_2_S were recorded continuously with documentation every minute. The mean and standard deviation (SD) of SaO_2_, rSO_2_S, and the FOES were calculated for every 30 min study period. Serial measurements of the variables were compared by ANOVA with repeated measures with 2 within-subject factors (before vs. after and bolus vs. continuous). A *p*-value < 0.05 was considered statistically significant.

## 3. Results

The study was conducted from October 2014 to April 2016. A total of 21 infants with a mean gestational age of 28.8 ± 1.6 weeks and a mean birth weight of 1260 ± 330 g were included. Age at the time of the study was 32.1 ± 13.6 days. All were on full enteral feeds of 157.71 ± 7 mL/kg/day (Table 1).

Thirteen study participants were diagnosed with respiratory distress syndrome (RDS) after birth (63%), eleven of whom received surfactant therapy (52%). Five infants (23%) had patent ductus arteriosus (PDA), three were treated with paracetamol, all were closed, and patients were asymptomatic. During the study, 50% of the participants were treated with caffeine, and 3 were still on oxygen supplementation. One infant developed NEC after completing the study and was treated conservatively. None of the study participants were diagnosed with significant IVH (grade 3–4) and none were diagnosed with BPD, defined as oxygen requirement at 36 weeks post-conceptional age.

Table 2 summarizes the results of the regional mesenteric and systemic saturation measurements during the study periods.

Our results demonstrate a significant decrease in rSO_2_S after the continuous feeding compared to the bolus feeding. In contrast, compared to continuous feeding, SaO_2_ decreased after bolus feeding. This finding was not statistically significant (Table 2).

The oxygen extraction by the intestinal tissue (FOES), which was calculated from the rSO_2_S and SaO_2_ data, was comparable between the two feeding regimens (Table 2).

## 4. Discussion

This prospective study assessed the effect of two different methods of enteral tube feeding on intestinal oxygenation and oxygen extraction among healthy, stable, appropriate-for-gestational-age preterm infants. We found that compared to a bolus feeding, rSO_2_S decreased after a continuous feeding, with no difference in the intestinal tissue oxygen extraction.

Previous studies that compared bolus to continuous feeding reported inconsistent results. Early studies that measured blood flow velocity in the superior mesenteric artery with Doppler demonstrated an increase in mesenteric blood flow 30–60 min after a meal [6,16,17]. The peak blood flow velocity was achieved faster when breast milk was given compared to formula [18].

The results of more recent studies that used NIRS technology also varied. Dave et al., showed that rSO_2_S, but not brain oxygenation, increased significantly an hour after feeding in stable preterm infants. However, continuous feeding was not assessed [9]. Demirel et al., used NIRS to examine the effects of positional changes on cerebral and mesenteric tissue oxygenation in very-low-birth-weight premature infants. They found no changes in regional cerebral or intestinal oxygenation with position change for an hour after bolus feeding [19].

In a subsequent study, Dave et al., compared bolus and continuous feeding using both NIRS and Doppler. They found an increase in rSO_2_S, as well as in blood flow velocity, in the superior mesenteric artery 30 min after bolus feeding among infants who were either average- or small-for-gestational-age. However, no change was demonstrated after continuous feeding. In agreement with our study, intestinal oxygen extraction was not affected by either feeding method in both study groups [14].

Corvaglia et al., also compared bolus and continuous feeding [15]. They measured cerebral regional oxygenation and rSO_2_S in 30 healthy, preterm infants during a bolus feeding and during a continuous feeding. Cerebral oxygenation was not affected by the method of feeding. Splanchnic oxygenation was increased following the bolus feeding and decreased after continuous feeding.

Many factors affect intestinal blood flow in preterm infants, including postnatal age, volume, and whether they receive human milk or formula. We assume that the increase in mesenteric blood flow after bolus feeding is mediated by the gastrointestinal hormones, gastrin, enteroglucagon, and gastric inhibitory peptide [20,21]. This hormonal response is probably not obtained following continuous feeding.

It is possible that we failed to find an increase in intestinal oxygenation after the bolus feeding due to a difference in feeding protocols. According to our local feeding protocol, a bolus feeding is given for 20–30 min, compared to 5–10 min in studies that did find the expected increased oxygenation after a bolus feeding.

Another possible cause is the duration of the measurements. In the current study, NIRS measurements were collected for 30 min after the end of the feeding. Although increased mesenteric blood flow was described previously during this period, others found it only after an hour [11].

Our study revealed a trend toward decreased systemic saturation after the bolus feeding that was not observed after the continuous feeding (*p* = 0.055). The clinical significance of this trend needs further evaluation. Previous studies investigated the influence of enteral feeding on systemic saturation. Desaturations were documented for bottle feeding, as well as tube feeding, regardless of the method of tube feeding [22]. However, while Toce and colleagues showed no change in the incidences of apnea during both bolus and continuous feeding [23], Akintorin et al., in agreement with our observation, found increased incidences of apneic events during bolus feeding [24].

The current study is limited primarily by the small sample size. A larger, follow-up study is needed to confirm the results and validate the conclusions. It is possible that a larger sample will reveal significant differences in systemic saturation and fractional oxygen extraction (FOES) as well. However, the main difficulty in recruiting patients for the study was the narrow window of opportunity between the achievement of full tube enteral feeding and the initiation of oral feeding. The study included only healthy, appropriate-for-gestational-age infants, on full enteral feeding. The results may not apply to unstable or small-for-gestational-age infants, who are at a higher risk for gastrointestinal complications. Another limitation is that all the participants received intermittent bolus feeding prior to enrollment. We cannot exclude the possibility of the results being somewhat influenced by the feeding method prior to enrollment in the study.

In conclusion, we demonstrated that intermittent bolus feeding and continuous feeding, two feeding techniques routinely used for preterm infants, affect regional intestinal oxygenation differently. We found a decrease in the rSO_2_S after continuous feeding and not after bolus feeding. On the other hand, there was a trend toward decreased systemic saturation after bolus feeding compared to continuous feeding.

The results may suggest that the mode of enteral feeding should be individualized and adjusted to the hemodynamic, respiratory, and nutritional status of each infant. Further studies should assess whether continuous feeding could be beneficial for infants coping with respiratory difficulties, while bolus feeding would be advantageous for those at a higher risk of hypoxic–ischemic gut damage (such as small-for-gestational-age infants).

To make useful recommendations regarding the best enteral feeding technique for preterm infants, it is important to include preterm infants during the first days of life, who tend to be less clinically stable.

## Figures and Tables

**Figure 1 children-09-00691-f001:**
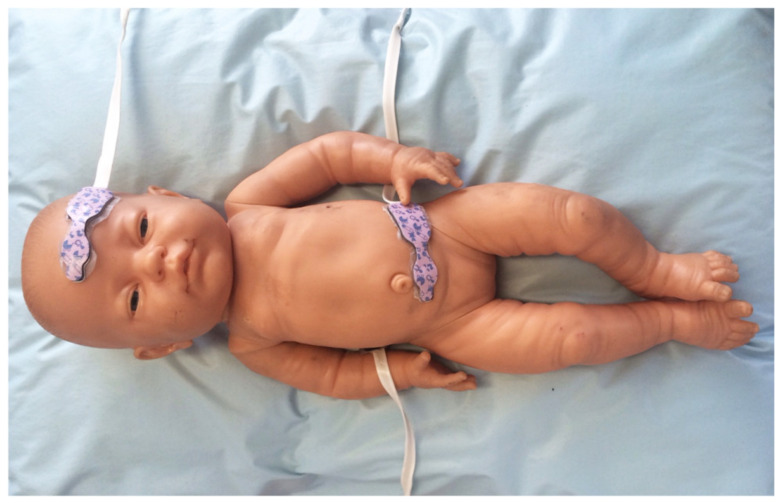
Near-Infrared Spectroscopy (NIRS) technology for assessment of mesenteric perfusion.

**Table 1 children-09-00691-t001:** Clinical characteristics of infants in the study group (N = 21).

Characteristic		Result
Gestational age at birth, weeks		28.81 ± 1.63
Birth weight, grams *		1270 ± 330
Male sex **		12 (57)
Cesarean section delivery **		15 (61)
Age at study, days *		32.05 ± 13.59
Gestational age at study, weeks		33.29 ± 1.38
Weight at study entry, grams *		1730 ± 310
Feeding volume at study entry, ml/kg/day *		157.71 ± 7
Gestational age at study entry, weeks *		30.61 ± 1.24
Type of feeding **	Fortified human milk	10 (48)
	Preterm formula	4 (19)
	Mix	7 (33)

* Mean ± SD; ** N (%).

**Table 2 children-09-00691-t002:** Changes in SaO_2_, rSO_2_S, and FOES during the 4 study periods.

Mean ± SD	Period 0	Period 1	Period 2	Period 3	* *p*
rSO_2_S	58.99 ± 10.71	58.78 ± 9.07	60.32 ± 10.02	55.60 ± 12.41	0.025
SaO_2_	97.43 ± 2.35	96.81 ± 2.50	97.29 ± 2.61	97.71 ± 1.73	0.055
FOES	0.39 ± 0.11	0.39 ± 0.09	0.40 ± 0.12	0.44 ± 0.11	0.129

* *p* refers to the difference between periods 0 and 1 compared to the difference between periods 2 and 3. Data presented as mean ± SD. rSO_2_S—Regional splanchnic oxygenation; FOES—fractional oxygen extraction ratio.

## Data Availability

The data presented in this study are available on request from the authors.

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
