# Peer review of "Regional Splanchnic Oxygenation during Continuous versus Bolus Feeding among Stable Preterm Infants"

_children, 2022, doi:10.3390/children9050691_

Round 1

Reviewer 1 Report

Gisela Laura Sirota and colleagues conducted this observational study to evaluate the effect of bolus versus continuous enteral feeding on regional splanchnic oxygen saturation and splanchnic fractional tissue oxygen extraction using the NIRS technique in preterm infants. The study is novel and well designed.

The following points have to be clarified:

  1. How did the authors make sure that the reading of NIRS belongs to splanchnic circulation? I mean using NIRS to monitor brain oxygenation has little variability since the brain is the only organ present in the skull. While in the abdomen other organs with different blood flow are present which may interfere with the purity of readings as being solely related to one organ.
  2. Did the authors correlate the NIRS readings with the Doppler assessment of splanchnic circulation to make a better correlation?
  3. Did the 5 infants with PDA receive medical therapy? Did included infants get an echocardiography assessment before enrollment in the study to check for PDA significance?
  4. In table 1 the authors stated that 8 infants received mixed formula and fortified human milk feeding. Feeding definitely is a factor that affects splanchnic circulation. Did the author ensure that the milk type was the same during the two interventions or did they leave the feeding type to whatever milk is available?
  5. The authors calculated a sample size of at least 12 infants required to detect a statistically significant difference of 10% in rSO2S 30 minutes after bolus feeding compared to continuous feeding, with 80% power at 0.05 levels. I did not get exactly how the author calculated their sample size? A 10 % reduction can be used to calculate the sample size of proportion outcomes while rSO2S is presented as the mean and standard deviation. I believe that the authors assumed a 10% difference between means. What were the means they used and what was their reference study for that? Also, is there a typo error regarding the calculated sample size? Should the sample in the methodology be written as 21 rather than 12?
  6. The Y-axis in the three figures better represents the unit of assessment (i.e. mean and SD, mean and CI, or median and quartiles, etc) with the two lines connecting the represented values before and after.
  7. What knowledge did the three figures add to the data presented in table 2?
  8. The authors stated that their results suggest that the mode of enteral feeding should be individualized and adjusted to each infant’s hemodynamic, respiratory, and nutritional status. It seems that continuous feeding could be beneficial for infants coping with respiratory difficulties, whereas bolus feeding may be important for those at higher risk of hypoxic-ischemic gut damage. I am not sure if the authors can come up with this assumption given that their study design did not test for different hemodynamic, respiratory, and nutritional statuses.

Author Response

Re: Regional splanchnic oxygenation during continuous versus bolus feeding among stable preterm infants

Gisela Laura Sirota, et al.

Reviewers’ comments

Reviewer 1

Q 1. How did the authors make sure that the reading of NIRS belongs to splanchnic circulation? I mean using NIRS to monitor brain oxygenation has little variability since the brain is the only organ present in the skull. While in the abdomen other organs with different blood flow are present which may interfere with the purity of readings as being solely related to one organ.

A.     The NIRS sensor was placed according to the manufacturer’s recommendations, under the umbilicus, where only intestinal tissue is close to the skin. The same location was used in previous studies. This sentence was added to the Methods section (revised manuscript, page 3, line 101).

Q 2. Did the authors correlate the NIRS readings with the Doppler assessment of splanchnic circulation to make a better correlation?

A.     We did not correlate NIRS readings with Doppler assessment; however, as indicated (Introduction, page 2, line 46), there was a good correlation between rSO2S and superior mesenteric artery blood flow measured by Doppler during feeding (references 11, 14 in the revised manuscript).

Q 3. Did the 5 infants with PDA receive medical therapy? Did included infants get an echocardiography assessment before enrollment in the study to check for PDA significance?

A.     We thank the reviewer for this comment and added the relevant information regarding PDA treatment and follow-up to the Results section. Five infants had patent ductus arteriosus (PDA), 3 were treated with paracetamol, all were closed and patients were asymptomatic at the time of the study (page 4, line 139).

Q 4. In table 1 the authors stated that 8 infants received mixed formula and fortified human milk feeding. Feeding definitely is a factor that affects splanchnic circulation. Did the author ensure that the milk type was the same during the two interventions or did they leave the feeding type to whatever milk is available?

A.     For the infants who were given a mixture of formula and fortified human milk, we made sure the same type of milk was given during both study periods (page 2, line 90).

Q5. The authors calculated a sample size of at least 12 infants required to detect a statistically significant difference of 10% in rSO2S 30 minutes after bolus feeding compared to continuous feeding, with 80% power at 0.05 levels. I did not get exactly how the author calculated their sample size? A 10 % reduction can be used to calculate the sample size of proportion outcomes while rSO2S is presented as the mean and standard deviation. I believe that the authors assumed a 10% difference between means. What were the means they used and what was their reference study for that? Also, is there a typo error regarding the calculated sample size? Should the sample in the methodology be written as 21 rather than 12?

A.     We rephrased the power analysis and added reference 14, which we relied on for this calculation. We included 20 infants in the study to cover for possible drop-outs (page 3, line 120).

Q 6. The Y-axis in the three figures better represents the unit of assessment (i.e. mean and SD, mean and CI, or median and quartiles, etc.) with the two lines connecting the represented values before and after.

A.     As Table 2 presents the same data as the figures, we accept your suggestions and deleted the figures.

Q 7. What knowledge did the three figures add to the data presented in table 2?

A.     As Table 2 presents the same data as the figures, we accept your suggestions and deleted the figures.  

Q 8. The authors stated that their results suggest that the mode of enteral feeding should be individualized and adjusted to each infant’s hemodynamic, respiratory, and nutritional status. It seems that continuous feeding could be beneficial for infants coping with respiratory difficulties, whereas bolus feeding may be important for those at higher risk of hypoxic-ischemic gut damage. I am not sure if the authors can come up with this assumption given that their study design did not test for different hemodynamic, respiratory, and nutritional statuses.

A. We agree with the reviewer and changed the manuscript accordingly (page 6, line 233).

Reviewer 2 Report

In this article by Sirota et al. entitled “Regional splanchnic oxygenation during continuous vs. bolus feeding among stable preterm infants” evaluated in 21 stable pre-term infant the regional splanchnic oxygenation (rSO2S) during continuous versus bolus feeding s.

The authors reported that rSO2S decreased significantly after continuous vs. bolus feeding (p=0.025), while there was a trend toward decreased SaO2 after bolus feeding (p=0.055). On the contrary, FOES, which reflects intestinal oxygen extraction, was not affected by feeding mode.

rSO2S is a challenging approach for the early identification of preterm infants at risk for necrotizing enterocolitis (NEC), given that it remains a significant, potentially fatal neonatal disease, and the study is of clinical importance.

However, I have some comments.

Abstract

  • The prospective nature of the study should also be stated in the abstract.
  • The statement that “Bolus feeding can potentially decrease the risk of hypoxic-ischemic injury to the intestines in preterm infants” remains to be proved, and thus should be deleted.

Introduction

  • Its already stated in the introduction that “only two small studies compared the influence of different methods of enteral feeding on intestinal oxygenation [14,15].” What does differentiate the earlier studies from the present? This should be clarified more precisely: for instance, a) the sample size (e.g., Corvaglia included 30 preterm infants), b) longer study period, c) the addition of FOES, d) different devices, … What’s new in this study?

Study protocol

  • It is written that “…the continuous feed was calculated as 25% of the total volume”. Why was that?
  • What are the SatO2 targets at your center? According to the values shown in table 2, some infants must have been given more oxygen during feeding (SatO2s of 96-97%, that is at or above the suggested upper limits). Also, could higher FiO2 and SatO2 may have affected the FOES values?
  • How were data (rSO2S, SatO2 and FOES) extracted to calculate means and SDs?

Results

  • Several neonatal problems (RDS, PDA, etc.) are reported in the study population. It would be good to know the medical problems at the time of the study.
  • Also, among the prematurity-related complication that were recorded, IVH and BPD were also included. Still, no results are given on these outcomes, not that BPD would have made any sense in the interpretation of the results in the present study.

Discussion

  • It should be noted that rSO2S (NIRS in general) is an indicator of oxygenation and perfusion.
  • A small comment on the accuracy of the Regional splanchnic oxygenation in neonates should be added. Regional Cerebral oxygenation is well accepted. Is this the case also with the splanchnic one?
  • How many infants during continuous feeding had the rSO2S decreased below the normal range, if there is any?
  • It is repeatedly mentioned that there was a trend toward decreased systemic saturation after the bolus feeding that was not observed after the continuous feeding. This may be true or not. However, I am really wondering what the clinical significance of such small change is; 97.4 to 96.8 with the bolus feeding.
  • Also, the authors suggest that “bolus feeding may be important for those at higher risk of hypoxic-ischemic gut damage”. This is a very vague term. Could they give some examples for clinical implementation?

Tables

  • The sum of neonates as regards the type of feeding (Table 1) is 22. However, the reported number of the studied neonates is 21.

Figures

  • I do not think that the figures are of additional value. Besides, the visualization of the magnitude in differences is a matter of the scale selected.

Author Response

Re: Regional splanchnic oxygenation during continuous versus bolus feeding among stable preterm infants

Gisela Laura Sirota, et al.

Reviewers’ comments

Reviewer 2

Q 1. Abstract: The prospective nature of the study should also be stated in the abstract.

A.     The prospective nature was added to the abstract (page 1, line 20).

Q 2. The statement that “Bolus feeding can potentially decrease the risk of hypoxic-ischemic injury to the intestines in preterm infants” remains to be proved, and thus should be deleted.

A.     We revised this sentence according to the reviewer's suggestion (page 1, line 31).

Q 3. Its already stated in the introduction that “only two small studies compared the influence of different methods of enteral feeding on intestinal oxygenation [14,15].” What does differentiate the earlier studies from the present? This should be clarified more precisely: for instance, a) the sample size (e.g., Corvaglia included 30 preterm infants), b) longer study period, c) the addition of FOES, d) different devices, … What’s new in this study?

A.      There is a difference in the feeding technique in our study compared to the previous studies. According to our local feeding protocol; a bolus feeding is given for 20-30 minutes, compared to 5-10 minutes in the previous studies and a continuous feeding is given during 5 hours, compared to 3 hours. The aim of our study was to examine its relevance to our feeding protocol. Actually, our results suggest that 5-10 minute feeding is more physiologic and further studies are needed.

Q 4. It is written that “…the continuous feed was calculated as 25% of the total volume”. Why was that?

A.      Based on our local feeding protocol, the volume of the continuous feeding is calculated by dividing the daily requirements (150 ml/kg/day) into 4 meals (25%). Each meal is given over a period of 5 hours and followed by an hour pause(page 3, line 93).

Q 5. What are the SatO2 targets at your center? According to the values shown in table 2, some infants must have been given more oxygen during feeding (SatO2s of 96-97%, that is at or above the suggested upper limits). Also, could higher FiO2 and SatO2 may have affected the FOES values?

A.     Our target saturation for preterm infants receiving oxygen is 90-95%. Most of the study participants (18/21) were on room air during the study and this explains their high SatO2 (page 4, line 142).

Q 6. How were data (rSO2S, SatO2 and FOES) extracted to calculate means and SDs?

A.      SaO2 and rSO2S were recorded continuously with documentation every minute. The mean and standard deviation (SD) of SaO2, rSO2S and FOES were calculated for every 30-minutes of the 4 study periods. This was clarified in the manuscript in the statistical analysis paragraph (page 3, line 123).

Q 7. Several neonatal problems (RDS, PDA, etc.) are reported in the study population. It would be good to know the medical problems at the time of the study.

A.     The average age of enrollment to the study was 32 days and the average corrected gestational age was 33 weeks (Table 1). Infants were on full feeds and clinically stable as per study protocol, only 3 were still on supplemental oxygen (page 4, line 139).

Q 8. Also, among the prematurity-related complication that were recorded, IVH and BPD were also included. Still, no results are given on these outcomes, not that BPD would have made any sense in the interpretation of the results in the present study.

A.     One infant had grade 1 IVH and another grade 2, none were diagnosed with significant IVH (grade 3-4) and none were diagnosed with BPD at 36 weeks. We added this information to the Results section (page 4, line 143).

Q 9. It should be noted that rSO2S (NIRS in general) is an indicator of oxygenation and perfusion

A.     Thank you for this remark. This was added to the Introduction (page 2, line 60).

Q 10. A small comment on the accuracy of the Regional splanchnic oxygenation in neonates should be added. Regional Cerebral oxygenation is well accepted. Is this the case also with the splanchnic one?

A. We agree that while cerebral NIRS is well accepted in several clinical indications (anesthesia, cardiac surgery and cardiopulmonary bypass), splanchnic NIRS is not routinely used. There is a good correlation between rSO2S and superior mesenteric artery blood flow measured by Doppler (ref 11). We believe that our study will support the use of NIRS to assess splanchnic oxygenation and perfusion, clinically.

Q 11. How many infants during continuous feeding had the rSO2S decreased below the normal range, if there is any?

A.     There is no normal range for rSO2S readings. It is used to track changes or trends in tissue perfusion and oxygenation. We found a decrease in the mean rSO2S after the continuous feeding in 14 of 21 patients.

Q 12. It is repeatedly mentioned that there was a trend toward decreased systemic saturation after the bolus feeding that was not observed after the continuous feeding. This may be true or not. However, I am really wondering what the clinical significance of such small change is; 97.4 to 96.8 with the bolus feeding.

A.     This minimal decrease in the saturation may not be clinically significant. Larger studies are needed. We added this reservation to the Discussion (page 5, line 203).

Q 13. Also, the authors suggest that “bolus feeding may be important for those at higher risk of hypoxic-ischemic gut damage”. This is a very vague term. Could they give some examples for clinical implementation?

A.     Our results suggest that bolus feeding could be advantageous for those at higher risk of hypoxic-ischemic gut damage, like small for gestational age infants. This example was added to the manuscript (page 6, line 232).

Q 14. The sum of neonates as regards the type of feeding (Table 1) is 22. However, the reported number of the studied neonates is 21.

A.      Thank you for your remark, 7 infants received mixed formula/fortified human milk and not 8. The table was corrected (page 3, line 135).

Q 15. Figures: I do not think that the figures are of additional value. Besides, the visualization of the magnitude in differences is a matter of the scale selected.

A.     We accept your suggestions and deleted the figures.

Round 2

Reviewer 1 Report

The authors sufficiently covered all inquiries and significantly improved the manuscript.

Reviewer 2 Report

None